# *In silico* screening of 393 mutants facilitates enzyme engineering of amidase activity in CalB

Martin R. Hediger[1], Luca De Vico[1], Julie B. Rannes[2], Christian Jäckel[2], Werner Besenmatter[2], Allan Svendsen[2] and Jan H. Jensen[1]

[1] Department of Chemistry, University of Copenhagen, Copenhagen, Denmark
[2] Novozymes A/S, Bagsværd, Denmark

## ABSTRACT

Our previously presented method for high throughput computational screening of mutant activity (*Hediger et al., 2012*) is benchmarked against experimentally measured amidase activity for 22 mutants of *Candida antarctica* lipase B (CalB). Using an appropriate cutoff criterion for the computed barriers, the qualitative activity of 15 out of 22 mutants is correctly predicted. The method identifies four of the six most active mutants with ≥3-fold wild type activity and seven out of the eight least active mutants with ≤0.5-fold wild type activity. The method is further used to screen all sterically possible (386) double-, triple- and quadruple-mutants constructed from the most active single mutants. Based on the benchmark test at least 20 new promising mutants are identified.

# INTRODUCTION

In industry, one frequently tries to modify an enzyme in order to enhance its functionality in a certain way (*Patkar et al., 1997*; *Kolkenbrock et al., 2006*; *Nakagawa et al., 2007*; *Naik et al., 2010*; *Takwa et al., 2011*). From an application point of view, one of the most interesting questions is how to modify an enzyme such that its activity is enhanced compared to wild type or such that a new kind of activity is introduced into the enzyme (*Jäckel, Kast & Hilvert, 2008*; *Frushicheva & Warshel, 2012*). It can therefore be of considerable relevance to have a method available which efficiently allows *a priori* discrimination between promising candidates for experimental study and mutants which can be excluded from the study.

Numerous methods are currently being proposed and developed for the description of enzyme activities, the theoretical background of which ranges from phenomenological and bioinformatics based approaches (*Chica, Doucet & Pelletier, 2005*; *Zanghellini et al., 2006*; *Zhou & Caflisch, 2010*; *Privett et al., 2012*; *Suplatov et al., 2012*) to quantum mechanics based *ab initio* descriptions (*Ishida & Kato, 2004*; *Noodleman et al., 2004*; *Friesner & Guallar, 2005*; *Rod & Ryde, 2005*; *Claeyssens et al., 2006*; *Hermann et al., 2009*; *Tian & Friesner, 2009*; *Parks et al., 2009*; *Altarsha et al., 2010*). However one can expect that methods which are highly demanding in terms of set-up efforts and computational time are less likely to be employed in industrial contexts where qualitative or semi-quantitative

Corresponding author
Jan H. Jensen, jhjensen@chem.ku.dk

**Figure 1 Reaction scheme for the formation of TI.** Nucleophilic attack by $O^\gamma$ of S105 on carbonyl carbon $C^{20}$ of substrate. $R_1$: $-CH_2-Cl$, $R_2$: $-CH_2-C_5H_6$.

conclusions can be of sufficient use in the beginning and planning phase of a wet-lab study. Few approaches, while taking into account a number of approximations and limitations in accuracy, aim at being used in parallel or prior to experimental work (*Himo, 2006*; *Hu et al., 2009*) and are not designed to be used for high throughput fashion.

Hediger et al. have recently published a computational method for high throughput computational screening of mutant activity (*Hediger et al., 2012*) and in this paper we benchmark the method against experimentally measured amidase activity for mutants of *Candida antarctica* lipase B (CalB) and apply the method to identify additional promising mutants.

## METHODS

We introduce the experimental set-up and the methodology for comparing experimental and computational data. We describe a benchmarking and a combinatorial study of CalB mutant activity.

Experimentally, variants of *Candida Antarctica* lipase B (CalB) were either produced in *Pichia pastoris* with C-terminal His6-tag for subsequent affinity purification or expressed in *Aspergillus oryzae* without terminal tag followed by a three-step purification procedure.

It is generally accepted that in serine protease like enzymes, the formation of the tetrahedral intermediate (**TI**, Fig. 1) is rate determining (*Ishida & Kato, 2003*; *Hedstrom, 2002*; *Fersht, 1985*; *Polgár, 1989*) and throughout this work we assume that a lower barrier for this reaction correlates to increased overall activity of the enzyme.

The substrate used throughout this study is N-benzyl-2-chloroacetamide. The organisms used for expression of the individual variants are indicated in Table 1.

### Generation of CalB variants without His-tags

Variants of CalB carrying the CalB signal peptide were generated at the DNA level using QuickChange mutagenesis on the corresponding gene residing in a dual *E. coli/Aspergillus Pichia pastoris* expression vector. The PCR was performed with proofreading DNA polymerase (New England Biolabs, NEB). To remove parent templates, they were methylated *in vitro* prior to PCR with CpG methyltransferase (from NEB) and digested *in vivo* after transformation of competent *E. coli* DH5 $\alpha$ cells (TaKaRa) according to the

**Table 1 Experimental overall activities and calculated reaction barriers of Set S.** Activity factors +1/−1 indicate increased/decreased overall activity. *Ao* and *Pp* indicating expression in organisms (Org.) *Aspergillus oryzae* or *Pichia pastoris*, respectively. The cutoff to distinguish higher and lower activity mutants is defined as 12.5 kcal/mol, see text.

| Species | Experimental | | Calculated | | Org. |
|---|---|---|---|---|---|
| | Activity [*WT] | Activity-Factor | Barriers [kcal/mol] | Activity-Factor | |
| G39A-T103G-W104F-L278A | 11.2 | 1 | 13.9 | −1 | *Ao* |
| G39A-L278A | 7.0 | 1 | 11.3 | 1 | *Pp* |
| G39A-W104F | 4.2 | 1 | 10.6 | 1 | *Ao* |
| G39A-T103G-L278A | 3.8 | 1 | 7.3 | 1 | *Ao* |
| G39A-W104F-L278A | 3.6 | 1 | 11.8 | 1 | *Pp* |
| T103G | 3.0 | 1 | 13.6 | −1 | *Ao* |
| G39A-W104F-I189Y-L278A | 2.9 | 1 | 10.9 | 1 | *Pp* |
| G39A | 2.8 | 1 | 11.2 | 1 | *Ao* |
| L278A | 2.5 | 1 | 12.8 | −1 | *Ao* |
| W104F | 2.0 | 1 | 12.0 | 1 | *Ao* |
| G39A-T103G-W104Q-L278A | 1.9 | 1 | 12.8 | −1 | *Ao* |
| G39A-T103G-W104F-D223G-L278A | 1.5 | 1 | 11.3 | 1 | *Pp* |
| G39A-T103G | 0.8 | −1 | 7.5 | 1 | *Ao* |
| G39A-T42A-T103G-W104F-L278A | 0.7 | −1 | 10.4 | 1 | *Pp* |
| I189H | 0.5 | −1 | 12.9 | −1 | *Pp* |
| G39A-I189G-L278A | 0.4 | −1 | 10.7 | 1 | *Pp* |
| G41S | 0.3 | −1 | 13.4 | −1 | *Pp* |
| I189G | 0.2 | −1 | 18.9 | −1 | *Pp* |
| G39A-T103G-W104F-I189H-D223G-L278A | 0.1 | −1 | 13.7 | −1 | *Pp* |
| G39A-T103G-W104F-I189H-L278A-A282G-I285A-V286A | 0.1 | −1 | 12.9 | −1 | *Pp* |
| A132N | 0.0 | −1 | 12.5 | −1 | *Pp* |
| P38H | 0.0 | −1 | 12.5 | −1 | *Pp* |
| | | | | | |
| WT | 1.0 | – | 7.5 | – | |

instructions from the manufacturer. Plasmid DNA was isolated from transformed *E. coli* strains, and sequenced to verify the presence of the desired substitutions. Confirmed plasmid variants were used to transform an *Aspergillus oryzae* strain that is negative in pyrG (orotidine-5′-phosphate decarboxylase), proteases pepC (a serine protease homologous to yscB), alp (an alkaline protease), NpI (a neutral metalloprotease I) to avoid degradation of the lipase variants during and after fermentation.

The transformed *Aspergillus* strains were fermented as submerged culture in shake flasks and the lipase variants secreted into the fermentation medium. After the fermentation, the lipase variants were purified from the sterile filtered fermentation medium in a 3 step procedure with (1) hydrophobic interaction chromatography on decylamine-agarose, (2) buffer exchange by gel filtration and (3) ion exchange chromatography with cation exchange on SP-sepharose at pH 4.5. The lipase variant solutions were stored frozen.

## Generation of CalB variants with His-tags

Variants of CalB carrying the CalB signal peptide and C-terminal His-tags were generated at the DNA level using SOE-PCR and inserted into a dual *E. coli*/*Pichia pastoris* expression vector using In-fusion cloning (ClonTech). The SOE-PCR was performed with Phusion DNA polymerase (NEB) and template DNA of the CalB gene. The cloned plasmids were transformed in competent *E. coli* DH5 $\alpha$ cells (TaKaRa). Plasmid DNA was isolated from transformed *E. coli* strains, and sequenced to verify the presence of the desired substitutions. Confirmed plasmid variants were used to transform a *Pichia pastoris* strain that is Mut(s), Suc(+), His(−). The transformed *Pichia* strains were fermented as submerged culture in deep well plates and secretion of the lipase variants into the fermentation medium was induced by the addition of methanol. After the fermentation, the lipase variants were purified from the cleared supernatants using a standard His-tag purification protocol (Qiagen) and buffer-exhanged into 50 mM phosphate buffer, pH 7.0, using Amicon Ultra centrifugal filter devices with a 10 kDa cutoff (Merck Millipore).

## Activity measurement

Amidase activity of CalB variants was determined in a two-step fluorimetric assay previously described by *Henke & Bornscheuer (2003)*. First, enzymatic hydrolysis of N-benzyl-2-chloroacetamide was performed in 96-well microtiter plates in 200 µL phosphate-buffered aqueous solution pH 7.0 including 10% organic co-solvent (THF or DMSO). Reactions containing 5 mM amide substrate, 0.3–3 µM enzyme, and 12 µg/mL BSA were incubated for 18–20 h at 37°C in a shaker incubator. In a second step, 50 µL of a 20 mM 4-nitro-7-chloro-benzo-2-oxa-1,3-diazole (NBD-Cl) solution in 1-hexanol was added and the reaction of NBD-Cl with benzylamine formed during amide hydrolysis proceeded under identical reaction conditions for another hour.

Fluorescence of the final reaction product was determined with excitation at 485 nm and measured emission at 538 nm. Calibration of the amide hydrolysis reaction was performed on each assay plate with benzylamine covering a concentration range between 0.05 and 5 mM. All enzymatic activities were corrected for non-enzymatic background reaction determined under identical conditions without enzymes present.

## Computational details

The computational method used to estimate the reaction barriers of the CalB mutants has been described in detail earlier (*Hediger et al., 2012*) and is only summarized here.

As described previously (*Hediger et al., 2012*), in order to make the method computationally feasible, relatively approximate treatments of the wave function, structural model, dynamics and reaction path are used. Given this and the automated setup of calculations, some inaccurate results will be unavoidable. However, the intent of the method is similar to experimental high throughput screens of enzyme activity where, for example, negative results may result from issues unrelated to the intrinsic activity of the enzyme such as imperfections in the activity assay, low expression yield, protein aggregation, etc. Just like its experimental counterpart our technique is intended to identify potentially interesting mutants for further study.

The reaction barriers are estimated computationally by preparing molecular model structures (*Hediger et al., 2012*) (consisting of around 840 atoms) of the enzyme substrate complex (**ES**) and the tetrahedral intermediate (**TI**) in between which linear interpolation is carried out to generate structures of the enzyme on the reaction path. Such adiabatic mapping is the most common way to estimate barriers in QM/MM studies of enzymatic reaction mechanisms. The resulting barriers tend to be in good agreement with experiment, which indicates that this is a reasonable approximation (see for example *Gao & Truhlar (2002)* and *Friesner & Guallar (2005)*). The geometry of each interpolation frame is optimized while keeping the distance between the nucleophilic carbon $C^{20}$ of the substrate and $O^{\gamma}$ of serine 105 (Fig. 1) fixed at a specific value $d_i = d_{ini} - i(d_{ini} - d_{fin})/10$, where $d_{ini}$ and $d_{fin}$ are the distances between $C^{20}$ and $O^{\gamma}$ in the **ES** complex and **TI**, respectively (in Å, 10 being the number of interpolation frames and $i$ the interpolation frame index). In geometry optimization calculations, the gradient convergence criteria is set to 0.5 kcal/(molÅ) and a linear scaling implementation of the PM6 method (MOZYME; *Stewart, 1996*) together with a NDDO cutoff of 15 Å is applied. The energy profile of the reaction barrier at the PM6 level of theory (*Stewart, 2007*) is subsequently mapped out by carrying out conventional SCF calculations of each optimized interpolation frame. All calculations are carried out using the MOPAC suite of programs (*Stewart, 1990*; *Stewart, 2009*). The molecular models are based on the crystal structure of the CalB enzyme with PDB identifier 1LBS (*Uppenberg et al., 1995*). In order to prevent significant rearrangement of hydrogen bonding network of surface residues during the optimization, a number of additional structural constraints are applied in the geometry optimizations, i.e., the residues S50, P133, Q156, L277 and P280 are kept fixed. These (surface) residues are observed to rearrange and form new hydrogen bonds in optimizations when no constraints are applied. Omitting the constraints leads to unconclusive barrier shapes containing many irregular minima along the reaction coordinate which do not permit to readily define a reaction barrier.

For the analysis, the reaction barrier is defined by the difference between the highest energy point on the reaction profile and the energy corresponding to the enzyme substrate complex. From our calculations (PM6/MOZYME in vacuum), we estimate the wild type (WT) barrier to be 7.5 kcal/mol.

Experimentally, specific activity of hydrolysis is determined. Given first order kinetics, saturation of the enzyme with substrate (usual for industrial application) and fast binding and product release, the catalytic rate constant $k_{cat}$ is directly proportional to the specific activity under the assumption that the amount of active enzyme remains constant. This therefore allows the catalytic rate constant $k_{cat}$ and, hence, the barrier height to be compared to the improvement factors reported in the results section. The approximations used here in relating the barrier height on the potential energy surface to $k_{cat}$ have been discussed previously (*Hediger et al., 2012*).

It is noted that using one CPU per interpolation frame on the reaction barrier, the complete barrier of one mutant can be computed with 10 CPUs usually within less than 12 h of wall clock time (for a molecular model of the size used in this study). Given a set of

**Table 2** **Point mutations.** The term *active site* refers to residues with potential direct Van der Waals contact to the substrate. The term *first shell/second shell* refers to residues which are adjacent to an active site/first shell residue.

| Target | Mutations | Type | Description |
| --- | --- | --- | --- |
| P38 | H | Second shell | (H neutral) |
| G39 | A | First shell | |
| G41 | S | First shell | |
| T42 | A | Second shell | |
| T103 | G | First shell | |
| W104 | F, Q, Y | Active site | |
| A132 | N | First shell | |
| A141 | N, Q | Active site | |
| I189 | A, G, H, N, Y | Active site | (G including additional water, H neutral) |
| D223 | G | First shell | (Increase of charge by +1) |
| L278 | A | Active site | |
| A282 | G | Active site | |
| I285 | A | Active site | |
| V286 | A | First shell | |

molecular models of the enzyme, and 100 available CPUs, it is possible to screen around 1000 mutants within one week.

## Combination mutants

The molecular model of the enzyme and the positions of the point mutations in the enzyme are illustrated in Fig. 2. The point mutations are listed in Table 2. Two sets of mutants are introduced in this section: a benchmarking set $S$ and a combinatorial set $L$, the definitions of which are provided in the following.

The point mutations are selected based on different design principles. These are either introduction of structural rearrangements in the active site to change the binding site properties of the active site (residues P38, G39, G41, T42, T103) (*Patkar et al., 1997*), introduction of space to accommodate the substrate (W104, L278, A282, I285, V286), introduction of dipolar interactions between the enzyme and the substrate (A132, A141, I189) (*Syrén et al., 2012*) or reduction of polarity in the active site (D223). Of course different heuristic considerations will apply for other enzymes when selecting the single mutations for combinatorial study. The mutants of the benchmarking study are collected in a small set $S$ (22 mutants, Table 1). For the combinatorial study, out of the above we select six residues (G39, T103, W104, A141, I189, L278) which, it is assumed, contribute strongest to increased activity and define the mutations at each position as listed in Table 3. Given the position $i$ and the number of mutations at each position $g_i$, in general the upper limit for the number of mutants $M$ in a combinatorial study can be calculated by writing a sum term for each type (i.e., "*order*") of combination mutant, i.e., single, double, ..., such

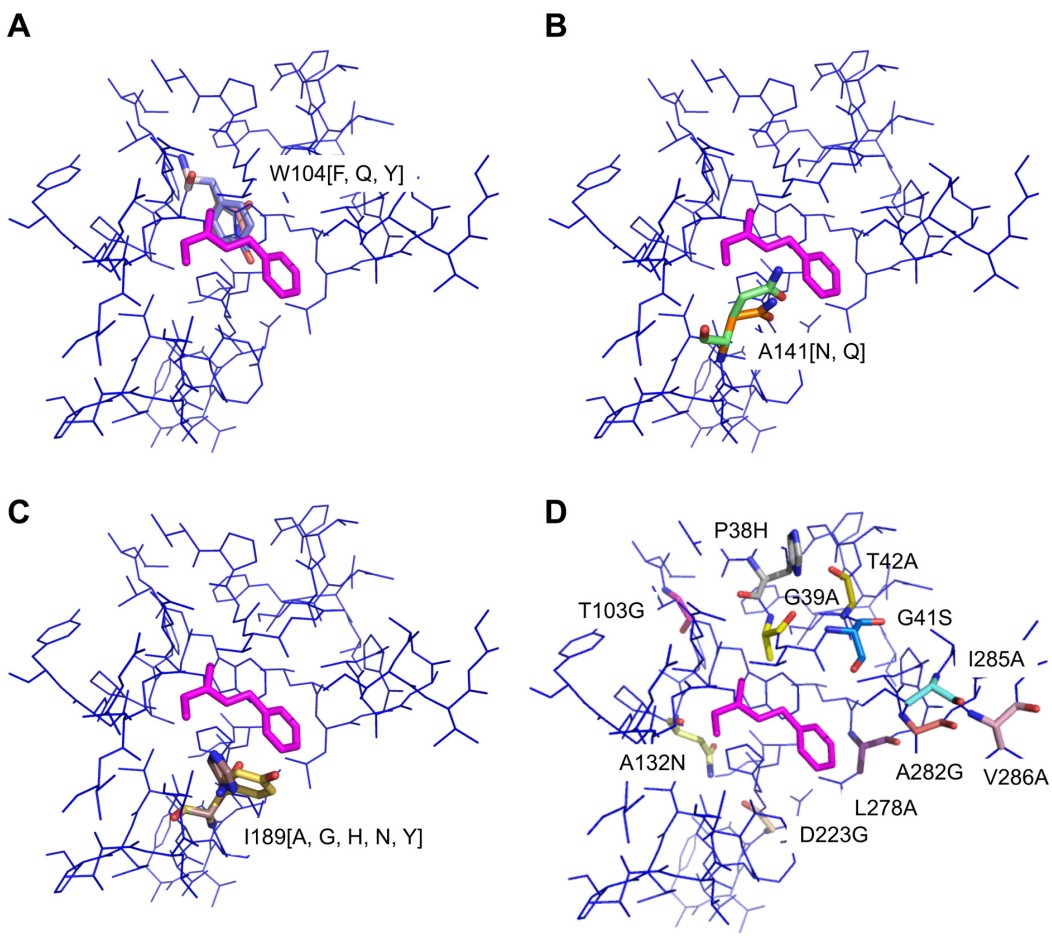

**Figure 2 Position of point mutations.** (A) Overlay of mutations W104[F, Q, Y]. (B) Overlay of mutations A141[N, Q]. (C) Overlay of mutations of I189[A, G, H, N, Y]. (D) Mutations P38H, G39A, G41S, T42A, T103G, A132N, L278A, A282G, I285A, V286A. Substrate shown in magenta.

that

$$M = \sum_i g_i + \underbrace{\sum_{\substack{i,j \\ j>i}} g_i \cdot g_j}_{\substack{\text{Double} \\ (o=2)}} + \underbrace{\sum_{\substack{i,j,k \\ k>j>i}} g_i \cdot g_j \cdot g_k}_{\substack{\text{Triple} \\ (o=3)}} + \cdots \tag{1}$$

$$\underbrace{\phantom{\sum_i g_i}}_{\substack{\text{Single} \\ (o=1)}}$$

where each sum term consists of $\binom{N}{o}$ individual terms ($N$ and $o$ being the number of positions which can be mutated and the order of the mutant, respectively). By this scheme, considering the mutations listed in Table 3, hypothetically 424 ($= 13 + 64 + 154 + 193$) single to four-fold mutants can be constructed. This number is reduced by applying the restriction that out of the 424 hypothetically possible mutants, 0 single, 2 double, 12 triple and 24 four-fold combination mutants including the pair A141N/Q-I189Y are discarded because in the molecular modeling, these side chains could not be allocated spatially in the same mutant. We further note that 15 out of these remaining 386 mutants (Table 3)

Table 3 **Side chains used for generation of combinatorial set *L*.** $i$ and $g_i$ indicate the position in the back bone and the number of mutations at that position, respectively.

| Mutation | $i$ | $g_i$ |
|---|---|---|
| G39A | 39 | 1 |
| T103G | 103 | 1 |
| W104{F, Q, Y} | 104 | 3 |
| A141{N, Q} | 141 | 2 |
| I189{A, G, H, N, Y} | 189 | 5 |
| L278A | 278 | 1 |

Table 4 **Combinatorial study details.** From the possible mutants, the combinations containing the pair A141N/Q-I189Y, the mutants with inconclusive barriers and the mutants with barriers >19.0 kcal/mol are subtracted to give the number of mutants in set *L*. "Only Set *L*" indicates the number of mutants uniquely present in set *L* and not in set *S*.

| Order | Possible | Containing A141N/Q-I189Y | Inconclusive barrier | Barrier > 19.0 [kcal/mol] | Set *L* | Only Set *L* |
|---|---|---|---|---|---|---|
| Single | 13 | 0 | 0 | 0 | 13 | 7 |
| Double | 64 | 2 | 4 | 8 | 50 | 47 |
| Triple | 154 | 12 | 21 | 20 | 101 | 98 |
| Four-fold | 193 | 24 | 36 | 19 | 114 | 111 |
| Total | 424 | 38 | 61 | 47 | 278 | 263 |

are present also in the benchmarking set *S* and thus the combinatorial study consists of 371 *unique* mutants. A detailed documentation of the number of screened residues in the combinatorial study is provided in Table 4.

Prior to analysis, the reaction barriers of the combination mutants are inspected visually and mutants with irregularly shaped barriers, i.e., consisting of multiple peaks of similar height along the reaction coordinate, are discarded. This step is done simply because the calculations yield inconclusive results, so the most conservative choice is to consider it a non-promising candidate for a more active variant. Generating plots of the profiles is completely automated and visual inspection can easily be done for hundreds of mutants. Furthermore, out of the mutants with regular reaction barrier shapes, we discard those mutants with barriers >19.0 kcal/mol (i.e., the largest calculated barrier from set *S*). Following these selection criteria, 61 mutants are discarded because of inconclusive barrier shapes and 47 mutants because the barrier is higher than 19 kcal/mol (a distribution of reaction barriers is shown in Fig. S1). After these filtering steps, 278 mutants remain in the combinatorial study which we collect in the large set *L* (out of which 15 are in set *S*). An overview on the distribution of reaction barriers for the mutants from set *L* is provided Fig. S2 of the supporting information.

We note that in set *S*, all barriers appear regular in shape and no mutant contains the A141N/Q and I189Y pair.

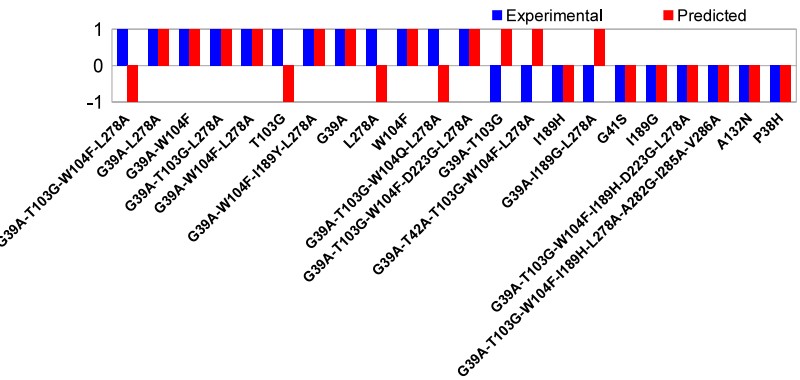

**Figure 3 Comparison of experimental and computed activities.** $1/-1$ correspond to increased/decreased overall activity, respectively. Prediction rate is 15/22(68%).

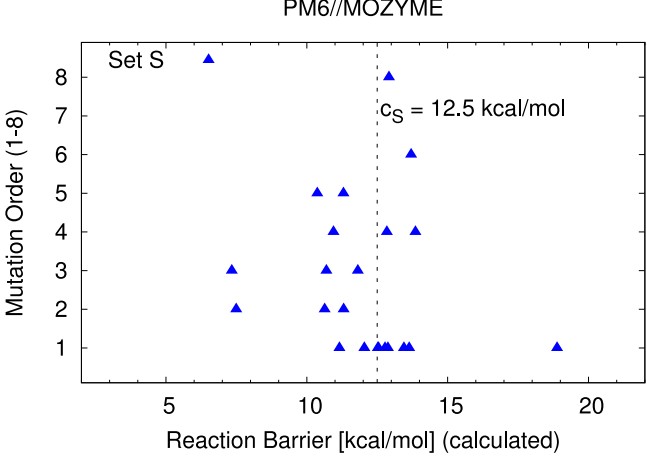

**Figure 4 Barrier scatter plot of set $S$.** 22 mutants; the cutoff value $c_S$ is discussed in the text.

## RESULTS AND DISCUSSION

### Set $S$: Calibration of the accuracy

The correspondence of the computed barriers from set $S$ with the experimental assay is shown in Fig. 3. The exact data is reported in Table 1. A scatterplot of calculated reaction barriers is presented in Fig. 4.

We note that in set $S$, the highest experimentally observed activity is around 11 times the wild type activity (G39A-T103G-W104F-L278A, Table 1), while roughly ten mutants show no increased activity. In total, six mutants show 3-fold or higher wild type activity. In the calculations, only one mutant is observed to have a lower barrier than the wild type (7.3 kcal/mol, G39A-T103G-L278A) and the highest observed barrier is 18.9 kcal/mol (I189G).

Given the approximations introduced to make the method sufficiently efficient, it is noted that the intent of the method is not a quantitative ranking of the reaction barriers, but to identify promising mutants for, and to eliminate non-promising mutants from, experimental consideration. Therefore only qualitative changes in overall activity are considered, which are represented by the activity factors ($+1/-1$).

We categorize the experimentally observed activities and the predicted reaction barriers as follows. From experiment, a mutant with activity of 1.2 (0.8) times the wild type activity or higher (lower) is considered as improving (degrading). Correspondingly, the computed difference in reaction barrier height between a mutant and the wild type is expressed in qualitative terms. For the comparison with the experimental activity assay, we define a barrier cutoff $c_S = 12.5$ kcal/mol to distinguish between *potentially* improving and degrading mutants in set $S$. The value of 12.5 kcal/mol is chosen such as to maximize the agreement with experiment, which is 68%, i.e., using a smaller or larger value for the cutoff will decrease this value.

A mutant with a predicted barrier $\geq c_S$ (12.5 kcal/mol) is considered to likely have decreased activity compared to the wild type while mutants with reaction barriers $< c_S$ are considered likely having increased activity.

We note that defining the cutoff is done purely for a *post hoc* comparison of experimental and computed data. When using the computed barriers to identify promising experimental mutants, one simply chooses the $N$ mutants with the lowest barriers, where $N$ is the number of mutants affordable to do experimentally (e.g., 20 in the discussion of set $L$).

Based on this approach, qualitative activity of 15 out of 22 mutants is correctly predicted. It is noted that the correlation is best for mutants with largest activity difference compared to wild type (both positive or negative). For example the method identifies four of the six most active mutants with $\geq$3-fold wild type activity. Similarly, the method identifies seven out of the eight least active mutants with $\leq$0.5-fold wild type activity. For mutants with only small differences in activity compared to wild type, the predictions are less accurate.

## Set *L*: Large scale screening study

Set $L$ is screened to identify new mutants for which increased activity is predicted. The 20 mutants with the lowest barriers are suggested as candidates for further experimental study in Table 5. The distributions of reaction barriers, resolved by mutations at positions 104 and 189, are shown in Figs. 5A and 5B.

In set $L$, three new mutants are identified with barriers lower than the predicted wild type barrier. Out of the 20 mutants suggested in Table 5, three are double mutants, seven are three-fold and ten are four-fold mutants. No single mutants were found for which increased activity compared to wild type is predicted. All mutants except one contain the G39A mutation, five contain the T103G mutation, six contain a mutation of W104, 13 contain a mutation of A141, 16 contain a mutation of I189 and eight contain the L278A mutation. From this observation it is likely that mutations of G39, A141 and I189 will likely

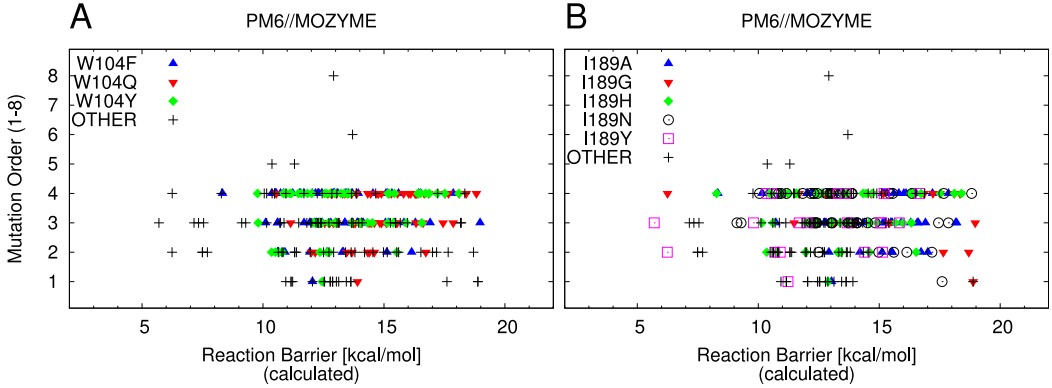

**Figure 5 Barrier scatter plots of set *L*.** In both panels, the labels indicate mutants containing the labeled and possibly additional mutations up to the indicated order. "OTHER" indicates a mutant not containing any of the labeled mutations or of higher than 4 order. (A) Mutations of W104. (B) Mutations of I189.

**Table 5 Selection of mutants from set *L* with lowest barriers.**

| Mutation | Barrier [kcal/mol] |
| --- | --- |
| G39A-T103G-I189Y | 5.7 |
| G39A-I189Y | 6.2 |
| G39A-A141Q-I189G-L278A | 6.3 |
| G39A-A141N-L278A | 7.6 |
| G39A-A141N | 7.7 |
| G39A-A141N-I189H-L278A | 8.3 |
| G39A-W104F-A141Q-I189A | 8.3 |
| G39A-A141Q-I189N | 9.1 |
| G39A-A141N-I189N | 9.3 |
| G39A-T103G-W104Y-A141N | 9.8 |
| G39A-W104Y-I189Y | 9.8 |
| G39A-A141N-I189N-L278A | 10.1 |
| G39A-W104F-A141N | 10.1 |
| G39A-I189H-L278A | 10.2 |
| G39A-A141N-I189A-L278A | 10.2 |
| W104Y-I189H | 10.4 |
| G39A-T103G-W104F-I189Y | 10.4 |
| G39A-A141Q-I189A-L278A | 10.4 |
| G39A-T103G-A141Q-I189H | 10.4 |
| G39A-T103G-I189A-L278A | 10.5 |

contribute to an increased activity of the mutant and should thus be included in future experimental activity assays.

Set *L* is further analysed in terms of the effect of the mutations at positions 104 and 189. For the mutations of W104, we note that single mutations which give rise to relatively high barriers (W104Q, W104Y, Fig. 5A) can have significantly lower barriers in combination with other mutations. For example, out of the sixty mutants with lowest barriers (Fig. S3),

33 contain a mutation of W104 out of which 17 are suggested to be W104F, while 14 are suggested to be W104Y (two contain W104Q).

The mutation of I189 is analysed in a similar way. In set *L*, five different mutations of this residue are screened (Table 3). The single mutant with the lowest barrier is I189Y and the two mutants with the lowest predicted barrier contain this mutation as well (Table 5). Similarly to above, higher order mutants containing I189A, I189G, I189H or I189N are predicted to have considerably lower barriers than the corresponding single mutants, Fig. 5B. Particularly, out of the mutants listed in Table 5, three contain the I189A, one contains I189G mutation, four contain the I189H mutation and three contain the I189N mutation.

As a special case we highlight that the single mutant I189G has one of the highest calculated barriers (18.9 kcal/mol, Table 1), however, the four-fold mutant G39A-A141Q-I189G-L278A has one of the lowest barriers (6.3 kcal/mol, Table 5). Interestingly, the mutant G39A-A141Q-L278A has an intermediate barrier (10.9 kcal/mol). It would appear that I189G as a single mutant is counterproductive (high computed barrier) but lowers the barrier of G39A-A141Q-L278A. This observation is further supported by the observation that the I189G mutation is in spatial proximity to A141Q. While it is difficult to quantify the interaction, it is likely that in the mutant, the rather large side chain of A141Q is better accommodated in the active site and can better interact with the substrate.

Observations such as these should be kept in mind when selecting the single mutants for consideration when preparing higher order mutants.

## CONCLUSIONS

Our previously presented method for high throughput computational screening of mutant activity (*Hediger et al., 2012*) is benchmarked against experimentally measured amidase activity for 22 mutants of *Candida antarctica* lipase B (CalB).

Experimentally, amidase activity is successfully introduced in 12 mutants, the highest activity is determined to be 11.2-fold over the wild type activity.

Using an appropriate cutoff criterion for the computed barriers, the qualitative activity of 15 out of 22 mutants is correctly predicted. It is noted that the correlation is best for mutants with largest activity difference compared to wild type (both positive and negative). For example the method identifies four of the six most active mutants with ≥3-fold wild type activity. Similarly, the method identifies seven out of the eight least active mutants with ≤0.5-fold wild type activity.

Thus validated, the computational method is used to screen all sterically possible (386) double-, triple- and quadrupole-mutants constructed from the most active single mutants. Based on the benchmark test at least 20 new promising mutants are identified.

These mutants have so far not been tested experimentally and are thus offered as scientifically testable predictions. Interestingly, we observe that single mutants that are predicted to have low activity appear to have high activity in combination with other mutants. This is illustrated in specific analysis of effects of mutations of two different positions (104 and 189).

### Funding

The work was funded by the EU through the *In Silico* Rational Engineering of Novel Enzymes (IRENE) project. The funders had no role in study design, data collection and analysis, decision to publish, or preparation of the manuscript.

### Grant Disclosures

The following grant information was disclosed by the authors:
EU through the *In Silico* Rational Engineering of Novel Enzymes (IRENE) project.

### Competing Interests

Julie B. Rannes, Christian Jackel, Werner Besenmatter and Allan Svendsen are employees of Novozymes A/S.

### Author Contributions

- Martin R. Hediger, Julie B. Rannes, Christian Jäckel and Werner Besenmatter conceived and designed the experiments, performed the experiments, analyzed the data, contributed reagents/materials/analysis tools, wrote the paper.
- Luca De Vico and Allan Svendsen conceived and designed the experiments, analyzed the data.
- Jan H. Jensen conceived and designed the experiments, analyzed the data, wrote the paper.

### Supplemental Information

Supplemental information for this article can be found online at http://dx.doi.org/10.7717/peerj.145.

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
