# Peer review of "In silico screening of 393 mutants facilitates enzyme engineering of amidase activity in CalB"

_PeerJ, doi:10.7717/peerj.145_

## Round 0.1 · original submission · Minor Revisions

· Academic Editor

Minor Revisions

Please address critical points raised by the first reviewer.

Reviewer 1 ·

Basic reporting

No Comments

Experimental design

No Comments

Validity of the findings

No Comments

Additional comments

This manuscript describes in silico screening of CalB mutants for enzymatic amidase activity. Then the computational results are benchmarked against experimental studies. The computational approach gives reasonable results in general -- the authors demonstrate that 15 out of 22 mutants are correctly predicted in qualitative activity. The manuscript is scholarly and well written. The methods are complete and clear. The data are new and appear to be true. I recommend publishing with minor revision.
I have a few minor suggestions/questions.
1. In table 1, two columns are both labeled with “Cat.” The authors need to make more detailed labels to distinguish (one from Exp. and the other is from Calc.). Also, the authors need to include the cutoff of 12.5 kcal/mol in the table legends otherwise it is difficult for readers to correlate calculated barriers with calculated cat.
2. In pg. 8, the authors state that the cutoff is done purely for a post hoc comparison of experimental and computed data. And they used 12.5 kcal/mol as cutoff. The authors need to expand this section to give detailed descriptions about how they reach the value of 12.5 in their study.
3. For results and discussion part, the two subtitles are “Set S” and “Set L”. It might be useful to guide readers if the authors can use more detailed subtitles which can generally summarize their results.
4. In the last part of results and discussion, the authors state that I189G as a single mutant is counterproductive but lowers the barrier of G39A-A141Q-L278A. This would suggest there are couplings between I189 and G39/A141/L278. While it is difficult to figure out such kind of couplings, one simple question is, is I189 close to G39/A141/L278 in the structure.

Reviewer 2 ·

Basic reporting

Meets all necessary criteria

Experimental design

Excellent

Validity of the findings

Excellent

Additional comments

This is an excellent paper, nicely demonstrating the utility of a practical computational modelling approach to the prediction of enzyme activity. The method uses semiempirical quantum chemical methods to model amidase reactivity in a lipase. This is thorough work of high quality. The paper is well written and the results are analysed and presented in appropriate detail. The results will be of wide interest. This is a demonstration of a method that will find real industrial application, as well as in other contexts. The paper is suitable for publication essentially as is. A minor point is that some of the references might be lacking information (e.g. refs 27 and 28 do not look complete as written). Also, the authors might want to comment a little more on how amidase activity is achieved. There is wide debate about amidase versus esterase activity, and, while this is not the focus here, comparison with enzymes such as fatty acid amide hydrolase, which has been the subject of much modelling work, could be useful (e.g. a little more discussion on what specific features may provide amidase activity and how the designed amidases compare to natural amidaases). This could assist in future rational design (as well as in testing the quantum chemical methods by comparisons on related reactions.

---

## Round 0.2 · accepted · Accept

· Academic Editor

Accept

Thank you for addressing all the critical points of the reviewers and for revising the manuscript accordingly.